# SpliceProt 2.0: A Sequence Repository of Human, Mouse, and Rat Proteoforms

**DOI:** 10.3390/ijms25021183

**Published:** 2024-01-18

**Authors:** Letícia Graziela Costa Santos, Vinícius da Silva Coutinho Parreira, Esdras Matheus Gomes da Silva, Marlon Dias Mariano Santos, Alexander da Franca Fernandes, Ana Gisele da Costa Neves-Ferreira, Paulo Costa Carvalho, Flávia Cristina de Paula Freitas, Fabio Passetti

**Affiliations:** 1Instituto Carlos Chagas, Fundação Oswaldo Cruz (FIOCRUZ), Rua Professor Algacyr Munhoz Mader 3775, Cidade Industrial De Curitiba, Curitiba 81310-020, PR, Brazil; 2Laboratory of Toxinology, Oswaldo Cruz Institute, Fundação Oswaldo Cruz (FIOCRUZ), Av. Brazil 4036, Campus Maré, Rio de Janeiro 21040-361, RJ, Brazil; 3Departamento de Genética e Evolução, Universidade Federal de São Carlos (UFSCar), Rodovia Washington Luis, Km 235, São Carlos 13565-905, SP, Brazil

**Keywords:** proteogenomics, transcriptome-informed protein databases, proteomics

## Abstract

SpliceProt 2.0 is a public proteogenomics database that aims to list the sequence of known proteins and potential new proteoforms in human, mouse, and rat proteomes. This updated repository provides an even broader range of computationally translated proteins and serves, for example, to aid with proteomic validation of splice variants absent from the reference UniProtKB/SwissProt database. We demonstrate the value of SpliceProt 2.0 to predict orthologous proteins between humans and murines based on transcript reconstruction, sequence annotation and detection at the transcriptome and proteome levels. In this release, the annotation data used in the reconstruction of transcripts based on the methodology of ternary matrices were acquired from new databases such as Ensembl, UniProt, and APPRIS. Another innovation implemented in the pipeline is the exclusion of transcripts predicted to be susceptible to degradation through the NMD pathway. Taken together, our repository and its applications represent a valuable resource for the proteogenomics community.

## 1. Introduction

Accompanying the rapid development of genomic technologies, protein high-throughput sequencing by mass spectrometry has made it possible to generate a continuous and large volume of data, fostering the development of new approaches to integrate and extract information from these data [1,2,3,4]. Proteogenomics integrates genomics, transcriptomics, and proteomics data to generate customized protein sequence repositories. Proteogenomics repositories can be tailored toward different conditions, such as samples, individuals, and habitats, based on the studied transcriptome, proteome, and genome information [1,2]. These customized databases are then used for mass spectrometry data searching, enabling the detection of unique peptides in each sample. This strategy may lead to a deeper understanding of several molecular processes such as gene expression, alternative splicing (AS), and protein synthesis. Moreover, the use of customized proteogenomics databases may improve the characterization accuracy of proteoforms and proteotypic peptides that are usually discarded when reference databases are used to analyze specific samples [5,6,7]. In short, customized proteogenomic databases open new opportunities for research in less explored areas.

One such area is AS, a molecular mechanism by which a gene can generate multiple transcripts, leading to more than one proteoform. Proteoforms are defined as different proteins encoded by the same gene, differing due to AS, nonsynonymous polymorphisms or post-translational modifications [8,9,10,11,12]. In humans, AS occurs in more than 90% of genes and constitutes an essential process of gene regulation [13,14,15,16,17]. AS is a conserved mechanism across species, and homologous genes in different species may be spliced in different ways, resulting in different proteins [18,19,20]. These genes may be classified as orthologous if they are shared by a species group, have a common ancestor, and show similar functions [21]. The similarity of the proteome of species whose genes are substantially similar, as is the case for humans and mice, has been evaluated in the context of AS [18,21].

AS is a molecular mechanism that strongly impacts the proteome and should not be overlooked when building optimized sequence repositories [18,20,22]. Errors and aberrations in AS may result in the development of numerous human diseases, including cancer, Alzheimer’s disease, Duchenne muscular dystrophy, and lateral amyotrophic sclerosis [13,14,15,23,24]. Faulty AS can result in the production of a truncated protein through the introduction of a premature termination code (PTC) in transcripts [25,26]. However, if this PTC is located between 50–55 nucleotides upstream of the last exon–exon junction, the transcript is likely to be degraded by the nonsense-mediated decay (NMD) pathway [27].

Most experimentally obtained peptides are associated with known protein sequences. Alternative transcripts may produce different polypeptide sequences, which may be absent from current reference proteome sequence repositories [12,28]. Therefore, the validation of transcripts and proteoforms created by AS is of utmost importance. Nevertheless, this remains challenging due to the need to detect proteotypic, proteoform-specific peptides by mass spectrometry [7,12,28,29,30]. To solve this issue efficiently, several customized databases of protein sequences and analysis pipelines have been developed, as available databases do not currently contain this information (e.g., Ensembl, RefSeq, UniProtKB/SwissProt, UniProtKB/TrEMBL, and NeXtProt) [31,32,33,34,35,36,37,38,39,40,41,42].

In this study, we present the latest release of SpliceProt [29], a customized proteogenomics repository focused on AS events using sequence and annotation data from Ensembl [28], UniProtKB/SwissProt [30,31], and APPRIS [32,33,34] to reconstruct transcript structures. Additionally, this new version includes the purging of computationally predicted NMD events [28,30,31,32,33,34]. Using our customized database, we improved proteotypic peptide identification through shotgun proteomics datasets from healthy human (*Homo sapiens*), mouse (*Mus musculus*), and rat (*Rattus norvegicus*) livers. We demonstrate the applications of SpliceProt 2.0 to explore proteotypic peptides, RNA-Seq data, and orthologous proteoforms. The complete pipeline and datasets are freely available at http://spliceprot.icc.fiocruz.br/ (accessed on 2 January 2024) [35].

## 2. Results

### 2.1. SpliceProt 2.0 Sequence Diversity at Transcript and Protein Levels

Table 1 describes the total amount of transcript variants identified by applying the ternary matrices methodology to transcript reconstruction, the number of variants selected for translation, and the polypeptide sequences obtained after in silico translation of all variants for human, mouse, and rat datasets.

### 2.2. Purging SpliceProt 2.0 Protein Sequences Predicted to Be Targeted for the NMD Pathway and First Methionine Sequence Selection

The NMD Classifier [36] predicted PTC-containing transcripts as targets of the NMD pathway in our dataset. The NMD Classifier also analyzes the possible AS event related to the insertion of PTC in the transcript (Figure 1). The NMD Classifier identified more NMD targets for the human dataset, with 16.50% NMD targets from the entire dataset, followed by 10.95% for mice and 1.45% for rats. We constructed SpliceProt 2.0 after purging proteins whose transcripts were predicted to be an NMD pathway target for each species. These datasets are available at http://spliceprot.icc.fiocruz.br/download.php (accessed on 2 January 2024).

In the development of SpliceProt’s latest release, we used a set of computational approaches to hypothetically translate all mRNA sequences (detailed description available in the Section 4). Following the in silico translation step of the transcripts, one method for evaluating the translation performed by the Transeq tool from the EMBOSS package (version 6.0) [37] based on ternary matrices was to compare such sequences with UniProtKB/SwissProt sequences. The results of this comparison revealed discrepancies in the initial methionine proposed in a subset of proteins that did not match with information from UniProtKB/SwissProt. Given this, and with the aim of being as conservative as possible in the choice of the initial methionine, considering its importance in translation, all SpliceProt sequences having corresponding Ensembl transcript IDs with the annotation made available in UniProtKB were modified to match the UniProtKB/SwissProt counterparts. This led to the modification of 2021, 813, and 114 sequences for human, mouse, and rat proteins, respectively, ensuring that the first methionine residue in SpliceProt, release 2.0 accurately reflected the UniProtKB/SwissProt data (Appendix A). Regarding the possible cause of these differences in the initial methionine, the very approach employed by Transeq in translating the three possible reading frames was a contributing factor.

### 2.3. The SpliceProt Release 2.0

This SpliceProt release 2.0 was generated using the Ensembl dataset aiming for an increase in the transcripts hypothetically translated and to permit the study of orthologous proteins among humans and two additional species, *Rattus norvegicus* and *Mus musculus.* The current version of SpliceProt relies on annotation data from Ensembl and APPRIS to reconstruct transcript structures, apply the removal of transcripts computationally predicted as susceptible to NMD events, and present a version for use in shotgun proteomics data analysis. The Splice 2.0 version for PSM search is the most recommended file for use as database search input files in shotgun proteomic data analysis, as the following modifications have been implemented in this dataset: (i) removal of entries whose sequences containing fewer than ten amino acids; (ii) labeling of sequences from SpliceProt 2.0 identical to those previously made available and manually annotated by UniProtKB/SwissProt; (iii) the marking and separation of sequences classified as canonical and variants based on the longest sequence; and (iv) the removal of sequences classified as susceptible to degradation through the NMD pathway by the NMD Classifier tool. Table 2 shows the number of protein sequences in each species classified as canonical and non-canonical according to the criteria presented in Section 4.1.

With a pairwise global sequence alignment approach, we compared the protein sequences generated in the current release of SpliceProt (2.0) with the protein sequences retrieved from OpenProt release 1.6 [38] and UniProtKB/SwissProt, as these are the databases most often used in proteogenomics analyses. We used the Ensembl transcript as the tracking identifier to point out protein sequences that were common to the various databases. Pairs of proteins sequences from distinct databases with no Ensembl ID correspondence were not compared.

The pairwise global sequence alignment comparison of SpliceProt 2.0 and SwissProt showed the highest identity levels among the protein sequences, followed by OpenProt 1.6 against SwissProt, and SpliceProt 2.0 against OpenProt 1.6 (Table 3 and Appendix A). The OpenProt is a proteogenomics repository that also employs the Transeq tool for the hypothetical translation of transcripts provided in other biological databases. However, OpenProt applies a cutoff value of only 30 codons to define the start of an open reading frame (ORF), in addition to considering multiple ORFs per transcript. Such a characteristic would reduce the identity sequence values in pairwise comparisons.

To investigate why the comparison between SpliceProt 2.0 and OpenProt yielded so many alignments with identities below 20% (Appendix A), we divided the dataset into four groups using the categorization defined by OpenProt as follows: the RefProt or reference proteins (known proteins annotated in RefSeq, Ensembl, and UniProtKB) were divided into two groups. The first group comprised all proteins annotated solely in RefSeq, while the second group included all proteins annotated by UniProtKB. The third group consisted of proteins categorized as New Isoforms (unannotated proteins with a significant sequence identity to a RefProt from the same gene). The fourth group comprised proteins categorized as AltProts (unannotated proteins with no significant identity to a RefProt from the same gene). An additional round of pairwise alignments was then performed to investigate this pattern (Appendix A). The results showed that most of the alignments with low identity and similarity values occurred when comparing sequences predicted by OpenProt as new Isoforms (Appendix A), mainly in the human datasets and in those from the RefSeq database in the human and mouse datasets (Appendix A). As expected for the proteins already annotated by UniProtKB/SwissProt, called *Reference proteins* by the OpenProt release 1.6, most alignments presented an identity value higher than the identity mean in both analyzed organisms (Appendix A). Transcripts without correspondence using the Ensembl ID between the two databases (Appendix A) were observed as classified and annotated by different consortia, such as RefSeq, UniProtKB/SwissProt, and OpenProt (categorized as new Isoforms and Alternative Proteins) (see Appendix A).

We also compared the SpliceProt 2.0 and OpenProt release 1.6 sequences to those provided by UniProtKB/SwissProt. The UniProtKB/SwissProt dataset was chosen as the reference in this analysis because its protein sequences are manually curated by experts [31]. Pairwise global sequence alignment revealed that SpliceProt 2.0 and SwissProt share 41,806, 22,651 and 4981 identical sequences for human, mouse, and rat, respectively (Appendix A). Pairwise global sequence alignment revealed that OpenProt 1.6 and SwissProt share 35,811, 20,942 and 7302 identical sequences for human, mouse, and rat, respectively (Appendix A).

The lower percentage of matches between SpliceProt 2.0 and the UniProtKB/SwissProt rat datasets is due to a bias in the number of sequences generated when the Ensembl version 100 dataset is used as input in the analysis. A set of the proteins and mRNAs annotated and revised by UniProtKB/SwissProt does not have a transcript with an equivalent Ensembl identifier. This reduced the number of possible comparisons among Ensembl transcripts [39] computationally translated in SpliceProt 2.0 and protein sequences from UniProtKB/SwissProt. Only 5161 identifiers had at least one ID directly associated between Ensembl [39] and UniProtKB/SwissProt annotation files.

The OpenProt is a proteogenomics repository that also employs the Transeq tool for the hypothetical translation of transcripts provided in other biological databases. However, they apply a cutoff value of only 30 codons to define the start of an open reading frame (ORF), in addition to considering multiple ORFs per transcript.

Although UniProtKB/TrEMBL contains significant information for proteogenomics analyses, we decided not to use UniProtKB/TrEMBL in these sequence comparisons at the global alignment level because not all UniProtKB/TrEMBL entries have a corresponding Ensembl transcript or gene identifier. The high levels of redundancies and errors in the annotations compromise the number of possible comparisons between repositories.

### 2.4. SpliceProt 2.0 Performance against Other Databases for Proteotypic Peptide Detection

The association of protein spectra to known protein sequences is still a challenge in the data analysis of shotgun proteomics. Transcriptomics data is currently accepted to validate findings obtained in PSM searches in shotgun proteomics, but its application is restricted to studies with both transcriptome and proteome data generated from the same organ or tissue. Another important application of transcriptomics is the generation of the repository of protein sequences, which will be used as a search file to generate the peptide-spectrum matches (PSMs) [43,44]. In mammals, the liver is the key organ in body homeostasis, and the role of AS in liver diseases and in healthy livers is poorly understood [45,46]. Therefore, we decided to use the liver as a model to investigate the PSM search performance in publicly available shotgun proteomics using the following databases: SpliceProt 2.0 version for PSM for the search, OpenProt release 1.6 [38,47] UniProtKB/SwissProt, and UniProtKB/TrEMBL. In Table 4, we present the contribution of each database in classical shotgun proteomics based on the criteria of Delta CN > 0.05 and primary score ≥ 2 [30,31,38,46,48,49].

Compared to OpenProt 1.6 [38], SpliceProt 2.0 generated 2 and 5 proteotypic peptides in common and 19 and 4393 unique proteotypic peptides using the human and rat repositories, respectively (Figure 2). No peptides in common were observed among the mouse repositories (Figure 2b). Peptides were considered proteotypic in this analysis if they were exclusive to a protein sequence in the database and identified strictly with SpliceProt 2.0.

Our analyses identified 21, 83, and 4393 unique proteotypic peptides that are not present in UniProtKB/SwissProt for human, mouse, and rat MS datasets, respectively (Figure 3).

Compared to UniProtKB/TrEMBL, 588, 1174, and 4393 proteotypic peptides were identified exclusively by SpliceProt 2.0 version for PSM search in human, mouse, and rat repositories, respectively (Figure 4).

Taken together, our results show that the number of redundant peptides between SpliceProt 2.0 and the other databases is always below five, especially for mice and rats (Figure 2, Figure 3 and Figure 4; Appendix A).

Identical proteotypic peptides identified among humans, mice, and rats were also compared. This comparison showed five proteotypic peptides in common among the results obtained (Figure 5).

### 2.5. Identification of Orthologous Proteoforms

The role played by AS events in evolution has not been fully clarified, and this has motivated our search for AS proteoforms shared between humans, mice, and rats [49]. The search for orthologous proteoforms between humans, mice, and rats identified 23,458 orthologous proteins between humans and mice, 13,633 between humans and rats, and 13,292 between mice and rats. Of these, 12,257 are orthologous between humans, mice, and rats (Table 5) (Appendix A).

A range of 14 to 18% of the proteoforms identified as being orthologous between (1) humans and mice, (2) mouse and rat, and (3) human and rat are identical. The remaining proteoforms (82–86%), also classified as orthologous by our approach, have pairwise identity scores between 60 and 99.9% (Figure 6). The number of identifications made separately by the Needle [37] and RBH [50] tools showed that the step performed by RBH resulted in the identification of two to 10 times more pairs of identical proteins than the Needle tool, suggesting that combining the tools considerably increases the number of predictions. For identifications with identity scores between 60 and 99.9%, the Needle tool [37] identified a larger number of pairs (Table 6).

The comparison between the results obtained from the analysis of publicly available healthy liver shotgun proteomics datasets and the list of orthologous proteoforms between humans, mice, and rats indicated 290 peptides identified in the liver proteomics datasets that are present in 23 orthologous proteins shared by all three species. These 23 proteins are supported by at least one peptide (Table 7) (see Appendix A). The primary score values denote high statistical confidence for all peptides (Table 8) (see Appendix A).

#### 2.5.1. Relationships between the Orthologous Proteoforms, Shotgun Proteomics Analysis, and Transcript Quantification

We used publicly available transcriptome data to confirm our results for the 23 proteins identified as orthologous between humans, mice, and rats that showed proteotypic peptides identified in the analysis of publicly available healthy liver shotgun proteomics datasets using our repository (Table 7). When analyzing transcripts whose hypothetical translation originated the orthologous proteoforms, we found 81 transcripts with TPM ≥ 1 (Table 7).

#### 2.5.2. Orthologous NMD Pathway Targets

We identified a set of three orthologous proteins in humans, mice, and rats that were predicted to be targets to the NMD pathway (Table 9). The NMD prediction provided by the NMD Classifier agreed with Ensembl annotations for all human transcripts and two mouse Ensembl transcripts.

### 2.6. Web Repository—User Interface

We present a simple and user-friendly tool to access SpliceProt 2.0 data (Figure 7a).

#### 2.6.1. Search Tab

The Search tab (Figure 7b,d) finds transcripts of a particular gene of interest. Users can enter a gene symbol in the query and select human, mouse, or rat species. They can also increase or decrease the graphical representation of the transcripts on the Chart Scale button. The gene search returns three main sections: (1) gene symbol, chromosome, strand, and genomic coordinates; (2) graphic representation of transcripts reconstructed by the ternary matrix methodology for the gene of interest; and (3) amino acid sequence obtained through the hypothetical translation of transcripts reconstructed by the ternary matrix methodology.

#### 2.6.2. Download Tab

In the Download tab (Figure 7c), users can obtain the following database versions of the SpliceProt 2.0 to use in classic analyses of raw shotgun proteomics data: (1) the entire sequence repository for humans, mice, and rats; (2) the repository, purged of NMD-predicted target sequences and sequences showing no redundancies with UniProtKB/SwissProt for humans, mice, and rats; (3) SpliceProt sequences computationally digested with trypsin and showing no redundancies with sequences from UniProtKB/SwissProt; and (4) the entire UniProtKB/SwissProt database used in our analysis.

#### 2.6.3. Submit Query Tab

The Submit Query tab (Figure 7e) is used to submit a FASTA file containing nucleotide sequences (cDNA) to a search for the sequences available in our repository. The search returns a FASTA file containing the protein obtained in the hypothetical translation of the submitted sequence to an e-mail address provided by the user.

## 3. Discussion

Proteogenomics studies and tools integrate data from genomics, mass spectrometry proteomics and transcriptomics to analyze sequence variations observed at the nucleotide and mRNA levels and their corresponding protein expressions [1,4,10,11,12,51]. However, the creation of new proteogenomics approaches and personalized databases is still a challenge for the proteomics community due to key factors, including: (i) the need to use a well-annotated reference transcriptome and a protein sequence file to perform the spectrum match with which to associate proteomics shotgun data; (ii) the need to integrate tools and results from different omics in a user-friendly way; and (iii) the characterization of non-canonical proteomes [1,4,52,53,54]. In this context, the characterization of non-canonical peptides is related to the identification of molecular mechanisms on the transcripts that can generate multiple proteins from the same gene, such as PTC and AS [18,20,25,27]. In this section, we discuss the results obtained in different applications of the SpliceProt 2.0 repository and present some evidence found in the literature about our findings and the role of the proteoforms presented as examples in both human and murines.

Comparisons performed at the global alignment level, and for the sequence identity and similarity scores between our repository, OpenProt 1.6, and UniProtKB/SwissProt, showed that the canonical sequences available in SpliceProt 2.0 are almost identical to the utilized version of UniProtKB/SwissProt from humans, mice, and rats. Based on this finding we removed identical sequences and peptides between SpliceProt 2.0 and UniProtKB/SwissProt to perform the identification of proteotypic peptides identified exclusively with SpliceProt 2.0 in the proteomics datasets.

Regarding sequence identity and similarity scores observed in the pairwise global alignments between SpliceProt 2.0 and OpenProt 1.6 using the Ensembl transcript ID to track hypothetic identical sequences, mean values in the human and mouse datasets did not reach 90%, indicating differences at the amino acid sequence level. This difference can be attributed to in silico translation, mainly from the computational reconstruction of transcripts using the ternary matrix approach [29] and coordinate selection filters such as TSL [55] and APPRIS [32,33,34] because both repositories use the Transeq tool [38] with different parameters to computationally translate cDNA sequences [29,38,47].

The results obtained by comparing the proteotypic peptides identified using the SpliceProt 2.0, OpenProt 1.6 and UniProtKB repositories indicate that SpliceProt 2.0 has an overall superior performance compared to OpenProt 1.6 in identifying unique proteotypic peptides in mouse and rat proteomics, particularly in the latter. Since rat genome annotation is precarious, the SpliceProt 2.0 repository of *Rattus norvegicus* has the potential to significantly contribute to the community. The qualitative results were another important feature observed when comparing the two repositories in the proteomics analysis.

The comparison between UniProtKB/SwissProt and SpliceProt 2.0 also indicates the potential of our repository to identify proteoforms and proteotypic peptides that are not represented in reference databases. The PatternLab V tool identified exclusive proteotypic peptides in the results obtained with our repository; nevertheless, important points about the use of large databases in the analysis of shotgun proteomics data should be raised. First, the size and scope of UniProtKB/TrEMBL can lead to an increased incidence of errors and redundancies in protein annotations. This is due to the inclusion of sequences automatically generated from large-scale sequencing data, which may contain assembly errors or inaccurate annotations [1,2,31,56,57,58,59,60,61,62]. Therefore, appropriate statistical parameters should be carefully chosen in proteomics shotgun data analysis tools, mainly in those affected by the size and curation of the database, such as FDR and significance thresholds [63,64,65,66,67].

The importance and possible roles of alternative splicing in the evolution, diversity and similarities present in two or more orthologous genomes is a hypothesis already raised by other authors such as Modrek and Lee (2003) [21] and Abril and colleagues (2005) in past decades [68]. The fact that the role of alternative splicing in evolution has not yet been fully clarified has motivated the search for alternative splicing proteins that are orthologous between human, mouse, and rat. Among our results, three examples of orthologous splice variants were selected that have multiple levels of support. The GLYCTK, NAXE, NUDT12, YWHAB and PSMC2 genes were chosen due to the similarity in both the number of exons and the architecture of the transcripts expressed in the three species (see Appendix A).

Glycerinate protein kinase (GLYCTK) is an enzyme found in the cytoplasm that has been identified in various species such as animals, plants, and bacteria. The structure of the glycerinate kinase genes is very similar between the three species in several aspects, such as the number and order of exons they contain (Appendix A). An interesting feature observed here involves the consequences of using the TSL as a parameter for choosing genomic coordinates in the hypothetical reconstruction and translation steps of a transcript, with two transcripts of a mouse Glyctk gene having a TSL value equal to 1 (ENSMUST0000036382 and ENSMUST112543) and another TSL value equal to 5, both producing the same protein according to our hypothetical translation. Another point to consider is the difference in the number of exons of human transcript GLYCTK-201 (ENST00000436784) reconstructed by our approach compared to the Ensembl transcript structure (Appendix A).

The NMD prediction by the NMD Classifier agrees with the Ensembl annotation for all those human transcripts and two mouse transcripts [37]. Although the limited number of identifications of orthologous proteins predicted to be targets of the NMD pathway, AP1S2 and FOXP3 have intriguing literature that should be further investigated in the context of the biological role of the NMD pathway regarding PTC mutations and human diseases [69,70,71,72,73].

Here, we present SpliceProt 2.0, the new release of a public proteogenomics database of known proteins and potential new proteoforms generated by the ternary matrices methodology [29]. SpliceProt 2.0 expanded the applications of SpliceProt repository as, besides human, also provides information for mouse and rat, model organisms for biomedical research. The value of SpliceProt 2.0 was demonstrated by the proteomic validation of splice variants absent from reference UniProtKB/SwissProt database, by its ability to predict orthologous proteins between human and murines (at both the transcriptome and proteome levels), and by providing evidence for AS-NMD-target conservation in mammals.

## 4. Materials and Methods

### 4.1. SpliceProt 2.0 Construction

The initial release of SpliceProt was based on annotations from UniGene for humans. Due to UniGene discontinuity [74], human data were updated to Ensembl version 100, and data for the mouse and rat species were included. Human and murine genomes (hg38/GRCh38, mm10/GRCm10, and Rnor 6.0) and transcriptome datasets (cDNA and ncRNA) were then obtained from the Ensembl Genome Browser, Version 100, to generate SpliceProt release 2.0. Transcript sequences were aligned with their respective genomes using the BLAT alignment tool [75]. The best alignments selected by pslReps [75] were ranked using an in-house Perl script, and only the best alignment coordinates were used as inputs for our pipeline [29].

In the previous version of SpliceProt, annotations from RefSeq [40] were used to support transcript structure quality control for external coordinates of the first and last exons (5′ and 3′ coordinates) before performing transcript reconstruction and the hypothetical in silico translation. By contrast, in the current release of SpliceProt, the Transcript Support Level (TSL) method from Ensembl was chosen to obtain only the transcripts with the highest degree of annotation reliability for humans and mice and the APPRIS system annotation [32,33,34] for rats. In the absence of the APPRIS and TSL flags, the transcript with the most extended sequence was chosen as the reference gene sequence to define the external coordinates used in the consensus table (see Appendix A).

#### 4.1.1. Computational Translation of Predicted Transcripts

The ternary matrix methodology was used to cluster similar splicing patterns for each Ensembl gene. In the ternary matrix methodology, each gene coordinate is represented by a matrix with three characters: “1”, “0” and “|”. Thus, each matrix row represents a transcript, and each column represents an exon or an intron. The exons are represented by “1”, introns by pipe characters “|” and “0” denotes exons in alternative spliced regions [29].

The hypothetical protein sequence was obtained for each cDNA sequence for a corresponding transcript using the Transeq program from the EMBOSS package [37], version 6.3.1. Transeq default parameters were selected to perform in silico translation in all three reading frames of the known gene orientation.

In the first stage, if the same splicing pattern represented more than one transcript, the most reliable transcript was identified to represent the variant and then submitted to the in silico translation step. Only transcripts annotated as protein-coding according to the manual annotation from the HAVANA project [43,55], lower TSL [55,76] and APPRIS values or longest sequence were selected. The extraction of gene consensus coordinates and selection of reading frames were performed as in the previous version of the repository [29].

A new step was introduced for the SpliceProt 2.0 predicted sequences to correct the methionine selected during the in silico translation step, as the first methionine was considered the correct point of translation initiation in the first SpliceProt version. If the first methionine of a SpliceProt translation does not match the first methionine present in the UniProtKB/SwissProt reference sequence, our sequence is corrected by removing the chosen methionine and subsequent amino acids until it matches the one from the UniProtKB/SwissProt using clustalw2 [77].

#### 4.1.2. Identification of Hypothetical Transcripts Predicted to Be Susceptible to Degradation by the Nonsense-Mediated Decay Pathway

Molecular events can generate transcript variants with no potential for translation, such as those sensitive to the nonsense-mediated decay (NMD) pathway [78,79]. Considering this characteristic, we needed to identify whether PTC was present in the transcripts used to input data when generating SpliceProt 2.0.

To this end, NMD Classifier was chosen to create a subset of our data without transcript variants predicted to be susceptible to the NMD pathway. NMD Classifier uses PTC distancing rules, adopting 50 nucleotides upstream of the last junction between exons or searching the annotation file to identify the transcript as a target of the NMD pathway. A customized GTF file based on the ternary matrix for each transcript and the coordinates of each gene available in SpliceProt 2.0 was created to perform the prediction of NMD pathway susceptibility. Additionally, we removed the “ENST” characters from the transcript name to force a de novo annotation for PTC by NMD Classifier. The NMD Classifier (Last update April 2019) analysis was performed at our GTF, but the software also requests a GTF from Ensembl (version 100) and a FASTA file for each chromosome sequence as input files (Ensembl, version 100).

### 4.2. Using SpliceProt 2.0 for Shotgun Proteomic Analysis

One of the main challenges in mass-spectrometry-based proteomics is to find the proteotypic peptides [80] with unique sequences in the database [81]. According to previous work [82], building customized databases based on combining complete AS canonical protein sequences and unique peptides of non-canonical proteins is a common approach in identifying proteotypic peptides. However, due to the large number of changes performed for the new release of our repository, after the hypothetical translation, the sequences were no longer submitted to a computational trypsin digestion step [29,37]. Instead, we prepared customized protein sequence databases for humans, mice, and rats, according to a novel strategy described below.

First, SpliceProt 2.0 sequences with less than 10 amino acids in the UniProtKB/SwissProt (available in 20 November 2021) were removed. A search was then performed for SpliceProt 2.0 sequences that are substrings of UniProtKB/SwissProt sequences and vice versa [31]. In both cases, only the UniProtKB/SwissProt sequence remained in the file. Finally, sequences from the SpliceProt 2.0 repository that showed 100% identity aligned to the UniProtKB/SwissProt sequences were removed from the SpliceProt 2.0 repository. Then, for each gene only unique sequences from the SpliceProt repository were selected and ordered by length, with the longest classified as canonical and the others as variants originating from the in silico translation of transcripts. At the end of this step, we generated the file corresponding to the SpliceProt 2.0 version that we argue to be optimized for PSM (peptide spectrum match) search (Figure 8). For genes that have a match with UniprotKB/SwissProt the canonical protein was elected by the definition of that database, that is indicated by “−1” after the UniProtKB/SwissProt ID in the FASTA file. All protein sequence databases are available at http://spliceprot.icc.fiocruz.br/download.php (accessed on 2 January 2024).

#### 4.2.1. Database Search Using Publicly Available Shotgun Proteomics Data

SpliceProt 2.0 version for PSM search, OpenProt 1.6, UniProtKB/TrEMBL and UniProtKB/SwissProt files, were used as database search input files to PatternLab V [63] using the default parameters and the carbamidomethylating of cysteine and oxidation of methionine modifications to be considered. The raw files were obtained from three studies that deposited raw data of MS1 and MS2 high-resolution MS/MS at ProteomeXchange [83], as follows: PXD008720 for humans, PXD020656 for mice, and PXD016793 for rats. We applied a 1% false discovery rate (FDR) at spectra, peptide, and protein levels to obtain a list of identifications in which we could have confidence. The common and default parameters used can be accessed in Appendix A.

#### 4.2.2. Proposed Strategy to Proteotypic Peptide Identification after Peptide Spectrum Match Search

The peptide spectrum match search is a widely adopted approach used to associate a given peptide spectrum to a protein sequence present in a database file. We needed to purge redundancies between SpliceProt 2.0 and UniProtKB/SwissProt to select the proteotypic peptides identified exclusively using the SpliceProt 2.0 version optimized for the PSM search file in the database search step using PatternLab V [63] (Figure 9a). The output files containing the identified peptide lists were filtered by applying three parameters:

1—Primary Score ≥ 2.5 and Delta CN > 0.05 [63] ;

2—Removal of redundant peptides in the same database with same MS1 values present in PatternLab V outputs. In these cases, we discard all duplicates and select only one peptide;

3—Removal of peptides identified in more than one protein sequence.

In the next step, the two lists of peptides returned by the applied filters, and consequently identified in only one protein, were compared (Figure 9b). Only peptides unique to SpliceProt 2.0 were selected for the next step (Figure 9c). After performing this last step, we obtained the list of proteotypic peptides identified exclusively by SpliceProt 2.0 (Figure 9c).

#### 4.2.3. Benchmarking for Classic Shotgun Proteomics Analysis Using Known Databases

We then performed a benchmarking-type analysis to verify similarities and differences between SpliceProt 2.0 and OpenProt 1.6, applied in the analysis of classic shotgun proteomics. The custom FASTA file indicated by OpenProt as its gold standard was used, with the following download parameters selected: AltProt and Isoforms, with a minimum of two unique peptides detected, and Ensembl Transcriptome Annotation [55]. We also used the datasets from UniProtKB (SwissProt and TrEMBL) [31] as they are widely used in classical shotgun proteomics analyses.

#### 4.2.4. SpliceProt, OpenProt, and UniProtKB Database Comparisons

To identify similarities and differences between the entries provided by our repository and OpenProt, release 1.6, since both repositories use the Transeq tool from the EMBOSS package [37], the hypothetical in silico translations from SpliceProt 2.0 were compared with those available in OpenProt, release 1.6, for human, mouse, and rat. The analysis was performed at the identity level between the sequences from each repository with the same Ensembl ID (ENST for human, ENSMUST for mouse, and ENSRNOT for rat), using the software Needle from EMBOSS package version 6.0 [37], with the parameters -gapopen 10.0 and -gapextend 0.5.

Considering UniProtKB/SwissProt as the reference database in annotation quality and protein sequence datasets, we also performed two other comparisons at the sequence identity level: SpliceProt 2.0 or OpenProt, release 1.6 versus UniProtKB/SwissProt. This analysis was also performed using the Needle tool [37], with the same parameters and criteria mentioned above.

The protein sequence global alignment comparative analysis between SpliceProt 2.0 and UniProtKB/TrEMBL was not performed. According to the literature, UniProtKB/TrEMBL still has limitations, lacking the full correspondence information with other bases such as Ensembl [39,55], resulting in the loss of data quality and reliability, which are important features in proteogenomics repositories [1,2,57,84]. 

### 4.3. Healthy Liver RNA-Seq Analysis

We performed a standard RNA-Seq analysis using public and controlled RNA-Seq data to analyze the expression of alternative splice variants in healthy human, mouse, and rat livers. Public RNA-Seq raw datasets were retrieved from the NCBI Gene Expression Omnibus with accession numbers GSE153986 (eight control samples) and GSE174535 (six wild-type samples). Human raw reads were retrieved from GTEx Consortium (phs000424) [84,85,86] (10 samples). The raw reads were trimmed with Illumina sequencing default parameters using TrimGalore software (version 0.6.6) [87]. We used HISAT2 (version 2.2.1.0) [88] to align the trimmed reads to human (UCSC hg38), mouse (UCSC mm10), and rat (UCSC Rnor 6.0) reference genomes. The SAM files were converted into BAM files and indexed using samtools, version 1.10 [89]. Salmon, v.0.12.0 [90] was used to quantify the number of reads mapped to each transcript. Additionally, Cufflinks, v.2.2.1 [91] was used to measure the relative abundance of transcripts, set with default parameters with genome annotation files (GTF format) available in Ensembl Project, version 102-104.

### 4.4. Identification of Orthologous Proteoforms

To identify orthologous proteoforms between humans, mice, and rats using the SpliceProt 2.0 repository, the available proteins were clustered based on the orthology gene classification available in Ensembl, release 100 [37] (Figure 10). Thus, three comparisons were performed: (1) human versus mouse, (2) human versus rat, and (3) rat versus mouse. For each pair of genes classified by Ensembl as orthologs, a new FASTA file was created containing the proteins encoded by the transcripts according to the predictions made in the in silico translation of the SpliceProt 2.0 FASTA file. From this step, grouped proteins were submitted to a pairwise alignment using the Needle tool from the EMBOSS package (version 6.0) [37] with default parameters (Figure 10).

Only protein pairs with alignments with similarity ≥ 80% and a gap value ≤ 0.1 were selected. Subsequently, protein pairs that were not selected in the previous step were subjected to the method described by Hernández-Salmerón and Moreno-Hagelsieb (2020) [50] to establish a new alignment coverage cutoff value using the Reciprocal Best Hit (RBH) value as a parameter to filter diamond [92] alignment outputs. In this step, protein pairs alignments with a target coverage ≥ 60% and query coverage ≥ 90% were selected (Figure 10).

Proteins were classified as orthologous between the three organisms when a perfect match between the pairs from the three analyses (human versus mouse, human versus rat, and rat versus mouse) was identified.

To investigate the possibility of finding cases in which orthologous proteins between humans, mice, and rats are susceptible to the NMD pathway, we applied the same pipeline above to proteins for which the respective transcripts were predicted as NMD targets.

### 4.5. Web Interface Implementation

All data and results obtained using the ternary matrix [29] and the hypothetical translation were deposited in a PostgreSQL database containing six tables: the first table stores all information from the alignment of cDNA/ncRNA to their respective genomes, such as coordinates, transcript identifiers, chromosome and identity scores; the second stores exon coordinates and identifiers; the third stores consensus coordinates; the fourth stores TSL or APPRIS values; the fifth stores gene symbols, protein sequences and transcript/protein identifiers; and the sixth stores ternary matrix outputs. SpliceProt 2.0′s web interface system was implemented using PHP, JavaScript, Canvaon in the front end, and Perl in the back end. We designed five principal sections (Figure 11): the ‘Home’ page describes SpliceProt 2.0 and guides users through the web repository; the ‘Search’ page allows for quick searches to be performed on a transcript and its hypothetical translation information using a gene of interest as input; the ‘Download’ page is used for downloading SpliceProt 2.0 datasets; the ‘Submit Query’ page serves to submit a FASTA file containing a nucleotide sequence and to discover the corresponding amino acid sequence obtained using the ternary matrix methodology; the ‘Contact’ page displays contact information to report bugs, feature requests, and other issues.

## 5. Conclusions

Here, we present the new and updated version of SpliceProt with several improvements. Using SpliceProt 2.0, we investigated orthologous relationships between proteoforms detected in humans, mice, and rats. Public shotgun proteomics data and RNA-Seq data from healthy human, mouse, and rat liver samples were analyzed to support our findings and demonstrate the usefulness of our sequence repositories.

Some of the main improvements implemented in this new release include: the inclusion of two new organisms in the repository (Mus musculus and Rattus norvegicus); the new source of cDNA sequences used as inputs to the ternary matrices methodology; the addition of two new annotation flags (TSL, APPRIS and HAVANA) to select reliable genomic coordinates for ternary representation construction; a new approach to creating a repository FASTA file for the analysis of shotgun proteomics data, making it possible to infer peptides missing in UniProt; and a description of how the user can identify orthologous proteoforms between humans and murines in our repository. Our web application also received upgrades, mainly in the back end with the development and implementation of new scripts in JavaScript and Perl. These new scripts decrease the response time of the results for end users of the application.

This methodology allowed us to analyze the latest version of the SpliceProt repository in terms of the numbers of identified proteoforms with evidence of proteotypic peptides and the relationship between expression levels in the analyzed transcriptome and proteome. Finally, we also used our repository to predict orthologous proteoforms between humans, mice, and rats. We correlated these findings with proteotypic peptides identified in these proteins by analyzing shotgun proteomic data. The annotation of proteoforms and splice junctions is not yet complete in biological databases such as RefSeq and UniProtKB/SwissProt, highlighting the need further to study the transcriptome’s diversity [93].

In this context, our results, obtained by analyzing shotgun proteomics data from the liver of three species, indicate that SpliceProt 2.0 may be used for the discovery of new proteoforms and disease biomarkers in humans and mice, with an improvement in the results for rats.

## Figures and Tables

**Figure 1 ijms-25-01183-f001:**
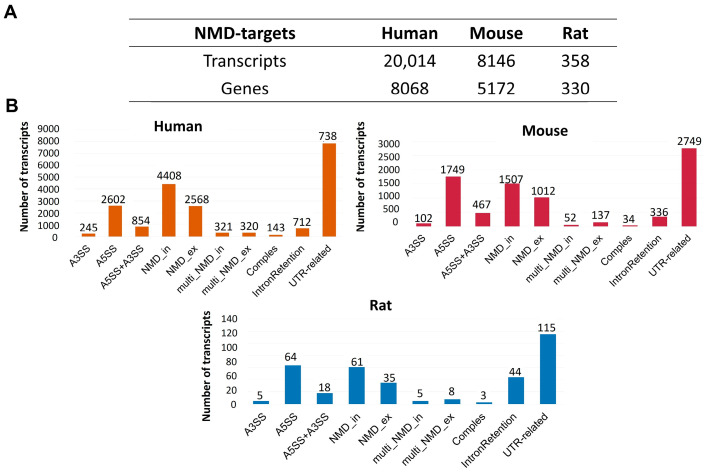
Descriptive statistics for SpliceProt2.0 predicted NMD-targets: (**A**) number of transcripts predicted as NMD targets and number of genes with at least one transcript predicted as an NMD target for each species at SpliceProt 2.0; (**B**) information about AS events associated with PTC insertion at NMD targets in each species. A3SS—alternative 3′ splice site, A5SS—alternative 5′ splice site, A3SS + A5SS—“both sites”, NMD_in—exon inclusion-caused, NMD_ex—exon exclusion-caused, multi_NMD_in—multiple exon inclusion-caused, multi_NMD_ex—multiple exon exclusion-caused, complex—“complex event”, IntronRetention—retention of an intron region and UTR-related—event associated with UTRs regions.

**Figure 2 ijms-25-01183-f002:**
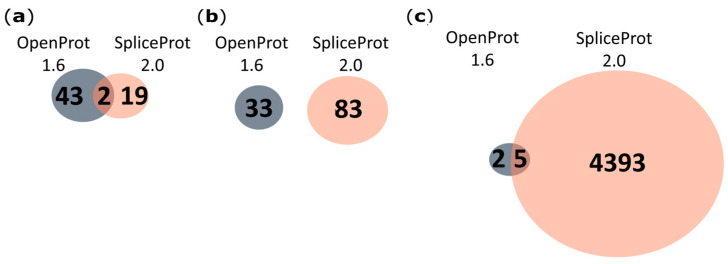
Venn diagram representing the comparison of proteotypic peptides identified with the PatternLab V tool using the SpliceProt 2.0 version for the PSM search (SpliceProt 2.0) and OpenProt 1.6 repositories: (**a**) human repositories comparison; (**b**) mouse repositories comparison; and (**c**) rat repositories comparison.

**Figure 3 ijms-25-01183-f003:**
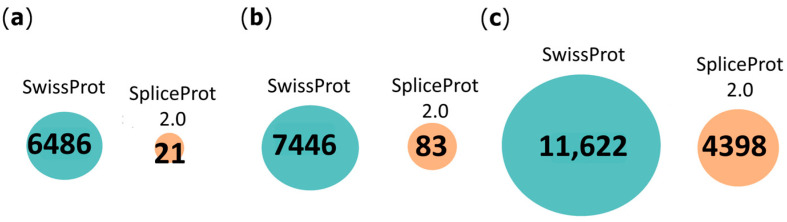
Venn diagram representing the comparison of proteotypic peptides identified with the PatternLab V tool using the SpliceProt 2.0 version for PSM search (SpliceProt 2.0) and UniProtKB/SwissProt (SwissProt) repositories: (**a**) human repositories comparison: (**b**) mouse repositories comparison; and (**c**) rat repositories comparison.

**Figure 4 ijms-25-01183-f004:**
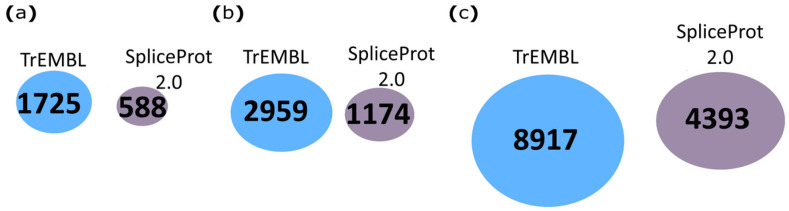
Venn diagram representing the comparison of proteotypic peptides identified with the PatternLab V tool using the SpliceProt 2.0 version for PSM search (SpliceProt 2.0) and UniProtKB/TrEMBL (TrEMBL) repositories: (**a**) human repositories comparison; (**b**) mouse repositories comparison; and (**c**) rat repositories comparison.

**Figure 5 ijms-25-01183-f005:**
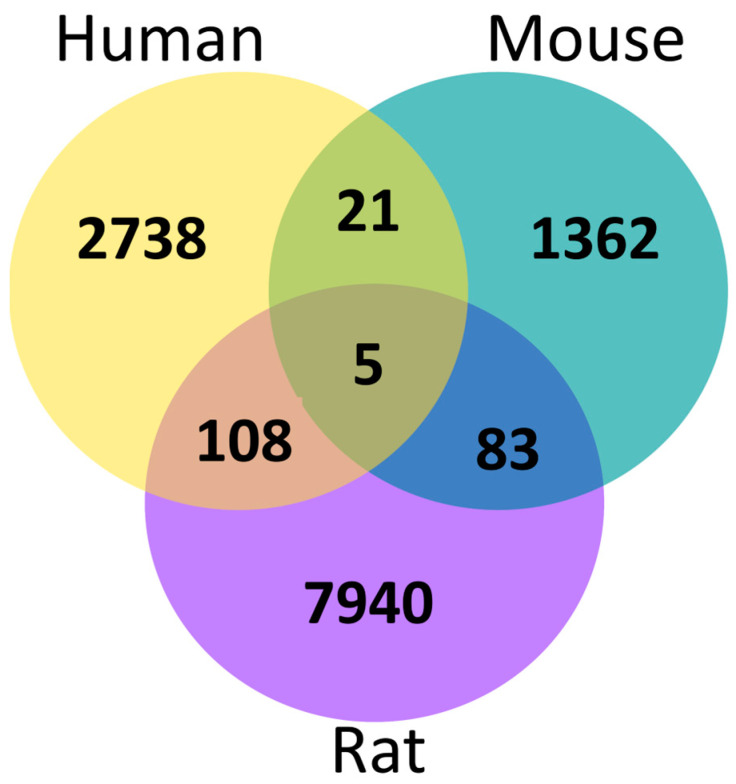
Venn diagram representing the comparison of identical proteotypic peptides identified by PatternLab V tool using SpliceProt 2.0 repositories for humans, mice, and rats.

**Figure 6 ijms-25-01183-f006:**
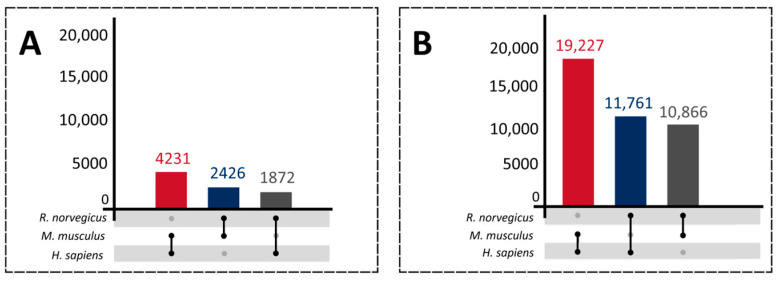
A UpSetR plot of orthologous proteoforms shared between humans, mice, and rats. The size of the intersections is shown as a bar chart above the matrix: (**A**) orthologous proteoforms with 100% identity: and (**B**) orthologous proteoforms with identity scores between 60–99.9%.

**Figure 7 ijms-25-01183-f007:**
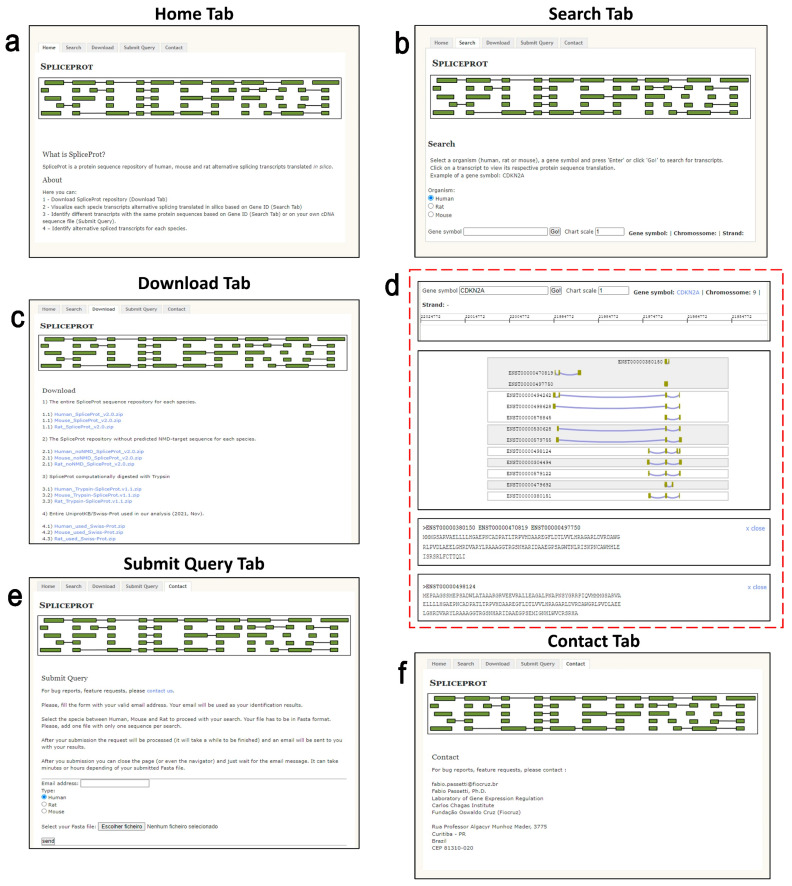
SpliceProt 2.0 web repository interface screenshots: (**a**) Home tab: description of SpliceProt 2.0 and brief user’s guide; (**b**) Search tab: search for gene of interest to see transcript and hypothetical translation information; (**c**) Download tab: page to download SpliceProt 2.0 datasets; (**d**) detailed search tab using CDKN2A as a gene of interest; (**e**) Submit Query Tab: tab to submit a FASTA file to receive the corresponding amino acid sequence in the user’s e-mail; and (**f**) Contact tab: contact information to report bugs, request features, and other issues.

**Figure 8 ijms-25-01183-f008:**
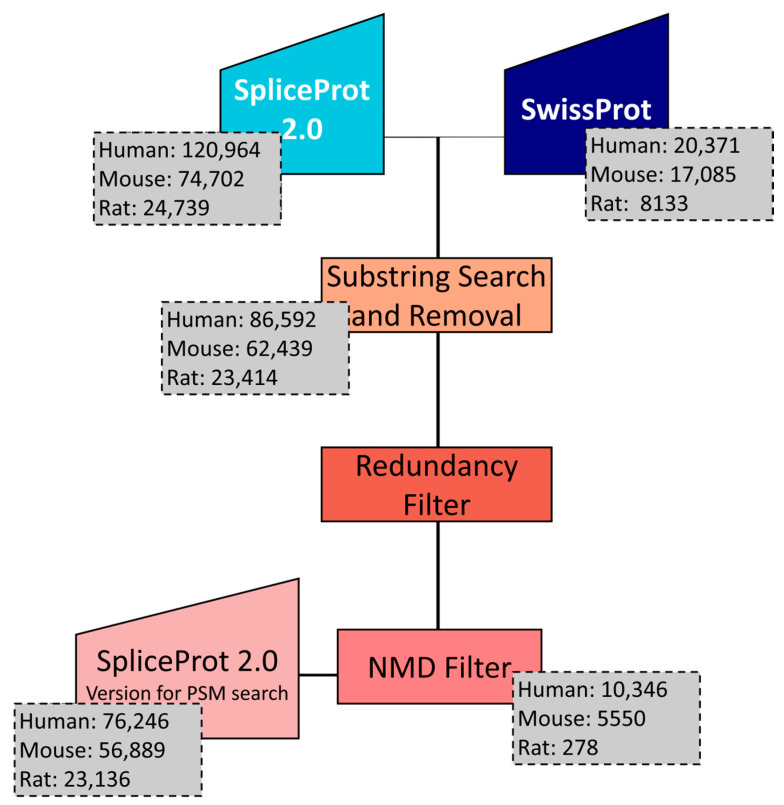
Flowchart exemplifying redundancy filters applied to obtain the SpliceProt 2.0 version for PSM search.

**Figure 9 ijms-25-01183-f009:**
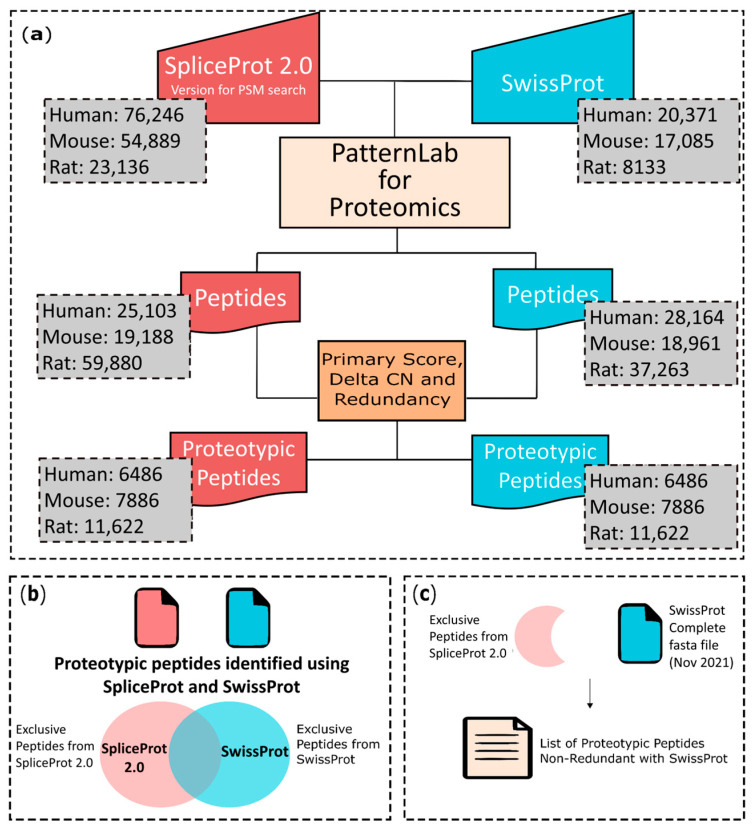
Flowchart showing the steps taken to filter the results obtained in the reanalysis of mass spectrometry data: (**a**) outputs from PatternLab V were compared and submitted to score cutoffs (Delta CN and Primary Score) and redundancy removal filters before reaching the identified proteotypic peptide level. The first gray boxes represent the number of entries from each database, FASTA files used in engine search for human, mouse, and rat SpliceProt 2.0 and UniProtKB/SwissProt portion repositories. The second gray box represents the number of peptides obtained in the raw PatternLab peptides and list outputs, and the number of proteotypic peptides are described in the third gray box; (**b**) identification of exclusive peptides from SpliceProt 2.0; and (**c**) comparison of exclusive peptides from SpliceProt to UniProtKB/SwissProt complete sequences to obtain a list containing only proteotypic peptides identified using SpliceProt 2.0 Full Version.

**Figure 10 ijms-25-01183-f010:**
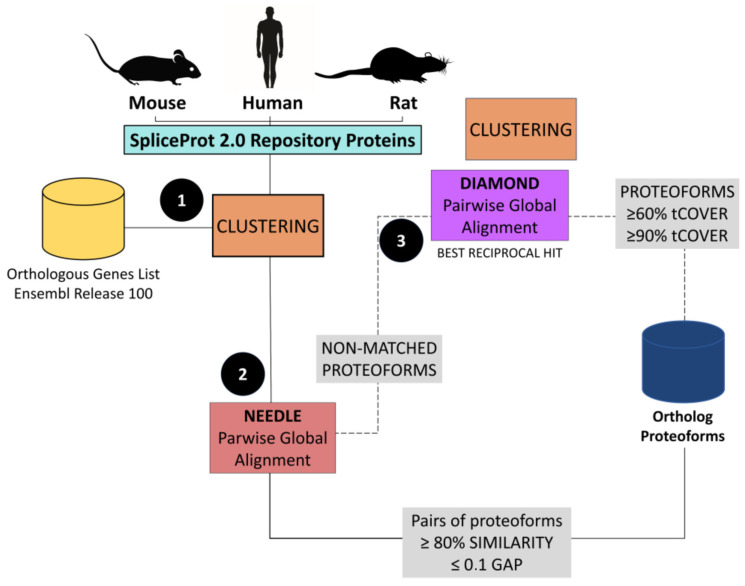
Workflow used to find ortholog proteoforms between humans, mice, and rats: (**1**) filter and clustering of proteins based on Ensembl Ortholog Genes List; (**2**) pairwise global alignment of human, mouse, and rat proteoforms pairs using Needle tool. Pairs of proteoforms with similarity ≥ 80% and gap ratio ≤ 1% were considered orthologs; and (**3**) proteoforms without a matched pair in the alignment from the Needle (Step (**2**)) were retrieved and subjected to Reciprocal Best Hit using the tool diamond. Pairs of proteoforms with target coverage ≥ 60% and query coverage ≥ 90% were considered orthologs.

**Figure 11 ijms-25-01183-f011:**
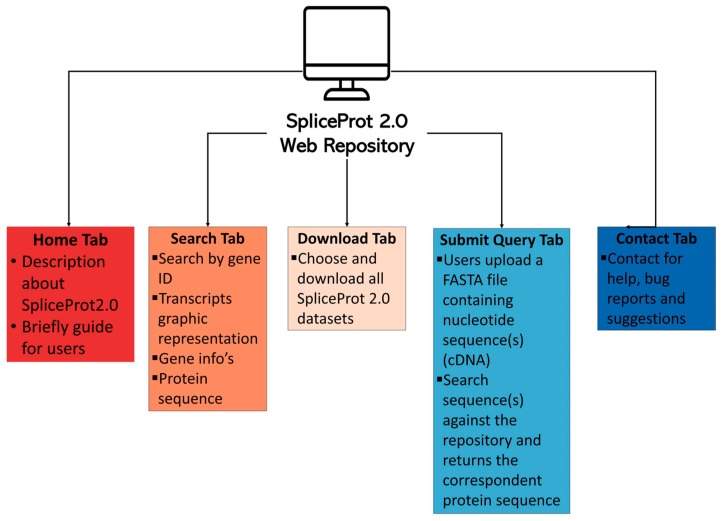
Architecture of SpliceProt 2.0 site.

**Table 1 ijms-25-01183-t001:** Total amount of transcript variants identified by ternary matrices methodology and of transcripts selected for computational protein sequence translation.

	Human	Mouse	Rat
Number of transcripts	242,578	135,694	37,453
Number of transcripts selected for computational translation	203,709	115,321	34,868
Number of polypeptide sequences obtained after computational translation	120,964	74,702	24,739

**Table 2 ijms-25-01183-t002:** Number of canonical and non-canonical proteins in the SpliceProt release 2.0.

	Canonical Proteins	Non-Canonical Proteins	Total Proteins
Human	30,976	55,616	86,592
Mouse	25,138	37,301	61,439
Rat	16,136	7251	23,387

**Table 3 ijms-25-01183-t003:** Mean and standard deviation from approximate identity values obtained in the pairwise global alignment of protein sequences retrieved from SpliceProt 2.0, SwissProt and OpenProt 1.6.

Species	SpliceProt 2.0 against SwissProt		OpenProt 1.6 against SwissProt		SpliceProt 2.0 against OpenProt 1.6	
	Mean	SD	Mean	SD	Mean	SD
Human	99.9	0.017	96.2	15.2	79.9	32.6
Mouse	99.9	0.018	98.2	9.36	87.3	26.0
Rat	99.9	0.009	96.2	15.2	97.0	11.0

**Table 4 ijms-25-01183-t004:** Number of peptides and proteins identified using liver shotgun proteomics studies [45,46] in each species for each database.

		SpliceProt 2.0 for PSM Search	OpenProt 1.6	UniProtKB/ SwissProt	UniProtKB/ TrEMBL
Human	Peptides	13,503	905	15,090	11,351
	Proteins	1805	451	1986	1243
Mouse	Peptides	10,375	437	9944	9560
	Proteins	1793	237	2347	1405
Rat	Peptides	20,400	122	13,504	15,021
	Proteins	4032	72	3212	3710

**Table 5 ijms-25-01183-t005:** Number of orthologous proteins identified.

	Datasets	Proteins
Total	human	120,932
	mouse	74,694
	rat	24,739
Orthologous	human/mouse	23,458
	human/rat	13,633
	rat/mouse	13,292
Triads (perfect match)	human/mouse/rat	12,257

**Table 6 ijms-25-01183-t006:** Quantitative number of pairs of proteoforms identified by Needle and RBH tools.

Comparison	Identity Score	Needle	RBH
human vs. mouse	100%	479	3752
human vs. rat	100%	203	2223
rat vs. mouse	100%	609	1263
human vs. mouse	60–99.9%	13,208	6019
human vs. rat	60–99.9%	7588	3278
rat vs. mouse	60–99.9%	10,180	1581

**Table 7 ijms-25-01183-t007:** The 23 proteins classified as orthologous between humans, mice, and rats with at least one proteotypic peptide as supporting evidence.

Human				Mouse				Rat			
Gene Symbol	Ensembl	Proteotypic Peptides	TPM	Gene Symbol	Ensembl	Proteotypic Peptides	TPM	Gene Symbol	Ensembl	Proteotypic Peptides	TPM
*ADK*	ENST00000539909	2	0	*Adk*	ENSMUST00000045376	2	145.1	*Adk*	ENSRNOT00000016709	1	156.8
*CMAS*	ENST00000229329	5	19.8	*Cmas*	ENSMUST00000032419	1	27.8	*Cmas*	ENSRNOT00000018734	1	49.3
*DDX3Y*	ENST00000336079	1	6.5	*Ddx3y*	ENSMUST00000091190	2	8.6	*Ddx3y*	ENSRNOT00000092078	2	0
*FAM120A*	ENST00000277165	3	20.2	*Fam120a*	ENSMUST00000060805	1	56.5	*Fam120a*	ENSRNOT00000060568	16	91.2
*FGA*	ENST00000651975	13	515.5	*Fga*	ENSMUST00000166581	8	62.6	*Fga*	ENSRNOT00000064091	15	2326.7
*GDI2*	ENST00000380191	3	58.7	*Gdi2*	ENSMUST00000223396	2	5.9	*Gdi2*	ENSRNOT00000024952	5	2703.4
*GLYCTK **	ENST00000436784	6	14.6	*Glyctk*	ENSMUST00000036382	2	52.3	*Glyctk **	ENSRNOT00000074595	3	211.5
*GLYCTK **	ENST00000436784	6	14.6	*Glyctk*	ENSMUST00000112543	2	34.5	*Glyctk **	ENSRNOT00000074595	3	211.5
*GLYCTK **	ENST00000436784	6	14.6	*Glyctk*	ENSMUST00000159809	2	8.2	*Glyctk **	ENSRNOT00000074595	3	211.5
*GLYCTK **	ENST00000436784	6	14.6	*Glyctk*	ENSMUST00000162562	2	21.2	*Glyctk **	ENSRNOT00000074595	3	211.5
*GPT2*	ENST00000340124	1	20.3	*Gpt2*	ENSMUST00000034136	1	186.2	*Gpt2*	ENSRNOT00000077275	3	100.1
*HSDL2*	ENST00000398805	4	25.2	*Hsdl2*	ENSMUST00000030078	1	33.6	*Hsdl2*	ENSRNOT00000059458	2	34.9
*HSPA4*	ENST00000304858	2	7.1	*Hspa4*	ENSMUST00000020630	6	33.5	*Hspa4*	ENSRNOT00000023628	7	34.8
*IQGAP2*	ENST00000274364	8	3.9	*Iqgap2*	ENSMUST00000068603	1	110.7	*Iqgap2*	ENSRNOT00000035017	38	101.5
*MTTP*	ENST00000265517	1	26.3	*Mttp*	ENSMUST00000029805	21	135.4	*Mttp*	ENSRNOT00000014631	4	186.2
*NAXE*	ENST00000368235	2	25.7	*Naxe*	ENSMUST00000029708	3	72.8	*Naxe*	ENSRNOT00000025986	2	47.8
*NUDT12*	ENST00000230792	2	4.5	*Nudt12*	ENSMUST00000025065	2	15.6	*Nudt12*	ENSRNOT00000066968	2	9.3
*PGRMC2*	ENST00000520121	3	0.2	*Pgrmc2*	ENSMUST00000058578	1	15.9	*Pgrmc2*	ENSRNOT00000018796	2	101.6
*PSMC2*	ENST00000292644	4	7.8	*Psmc2 **	ENSMUST00000030769	2	52.4	*Psmc2 **	ENSRNOT00000016450	4	79.9
*PSMC2*	ENST00000425206	4	7	*Psmc2 **	ENSMUST00000030769	2	52.4	*Psmc2 **	ENSRNOT00000016450	4	79.9
*PSMC2*	ENST00000435765	4	0	*Psmc2 **	ENSMUST00000030769	2	52.4	*Psmc2 **	ENSRNOT00000016450	4	79.9
*PSMD1*	ENST00000308696	4	13.8	*Psmd1*	ENSMUST00000027432	1	54.8	*Psmd1*	ENSRNOT00000024306	1	64.5
*SCFD1*	ENST00000458591	4	5.7	*Scfd1*	ENSMUST00000021335	3	28	*Scfd1*	ENSRNOT00000040548	2	23.9
*SEC24D*	ENST00000280551	1	11.6	*Sec24d*	ENSMUST00000047923	1	26.9	*Sec24d*	ENSRNOT00000064809	8	38.5
*STIP1*	ENST00000305218	1	31.6	*Stip1*	ENSMUST00000025918	1	37.3	*Stip1*	ENSRNOT00000028743	4	56.8
*UGT1A1*	ENST00000305208	5	203.8	*Ugt1a1*	ENSMUST00000073049	3	508.8	*Ugt1a3*	ENSRNOT00000025045	3	211.2
*XYLB*	ENST00000207870	1	6.2	*Xylb*	ENSMUST00000039610	1	19.7	*Xylb*	ENSRNOT00000019106	6	56.3
*YWHAB*	ENST00000353703	4	25.3	*Ywhab*	ENSMUST00000018470	6	27.2	*Ywhab **	ENSRNOT00000016981	6	30.7
*YWHAB*	ENST00000372839	4	3.7	*Ywhab*	ENSMUST00000131288	6	0.4	*Ywhab **	ENSRNOT00000016981	6	30.7

* Orthologous protein comes from more than one transcript according to the hypothetical translation based on the reconstruction of transcript ternary matrices.

**Table 8 ijms-25-01183-t008:** List of identified peptides of the proteins from Glycerate Kinase gene.

Peptide	Primary Score	Human	Mouse	Rat
AVLGMAAAAEELLGQHLVQGVISVPK	5.76	X	-	-
LLAARGATIQELNTIRK	4.77	-	X	-
ADSDPHGPHTCGHVLNVIIGSNSLALAEAQR	4.88	-	-	X
GPVCLLAGGEPTVQLQGSGK	4.03	-	X	X
GPVCLLAGGEPTVQLQGSGK	4.45	-	X	X
GPVCLLAGGEPTVQLQGSGR	3.72	X	-	-

**Table 9 ijms-25-01183-t009:** Orthologous NMD targets.

Human		Mouse		Rat	
Gene Name	Ensembl ID	Gene Name	Ensembl ID	Gene Name	Ensembl ID
*PLCB4*	ENST00000492632	*Plcb4*	ENSMUST00000184371	*Plcb4*	ENSRNOT00000049855
*AP1S2*	ENST00000672063	*Ap1s2*	ENSMUST00000140845	*Ap1s2*	ENSRNOT00000081652
*FOXP3*	ENST00000651307	*Foxp3*	ENSMUST00000234479	*Foxp3*	ENSRNOT00000091146

## Data Availability

All materials, including datasets, pipelines, and analysis notebooks, are available at https://github.com/grazLet/spliceprot, accessed on 1 December 2023. Human, mouse, and liver datasets used in this manuscript include RNA-seq data (dbGaP Accession phs000424.v8.p2, Gene Expression Omnibus GSE174535, GSE153986) and mass spectrometry data (ProteomeXchange: PXD008720, PXD020656, PXD016793). The dataset(s) supporting the conclusions of this article is(are) available in the SpliceProt 2.0 repository, [http://spliceprot.icc.fiocruz.br]. accessed on 1 December 2023. The dataset(s) supporting the conclusions of this article are included within the article (and its Appendix A).

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
