# Peer review of "SpliceProt 2.0: A Sequence Repository of Human, Mouse, and Rat Proteoforms"

_ijms, 2024, doi:10.3390/ijms25021183_

Round 1

Reviewer 1 Report

Comments and Suggestions for Authors

This manuscript updated their previously developed SpliceProt database which provide more proteforms predictions not only in human but also in rat and mouse. Overall, it’s good. However, please provide high-resolution figures.

This manuscript updated their previously developed SpliceProt database by including additional mouse proteome.
1. this study aims to list the proteins in human and mouse. However, there are already many websites such as human and mice databases that can precisely list the known proteins and genes. in the result 2.2, your predictions of proteins with 1st methionine is different from Uniprot database, and you corrected it to follow the uniport. I may have a little doubt about accuracy in your new release.
2.  in the result 2.3, paragraph 4, the first sentence, SpliceProt 2.0 and OpenProt have many alignments with identity below 20%, I didn’t see any data, table 3?
3. In table 2, 3, 4, there are so many different peptides you predicted when using your own SpliceProt, is it accurate.  
4. the whole context seems redundant, hard to understand, express more succinctly please.
5. For the reference part, provide less papers and more relevant references! Don’t put the same reference in so many places.
6. Keep all the figures the same fonts, and provide high resolution figures, we can’t see clearly.

Reviewer 2 Report

Comments and Suggestions for Authors

The font in figure 1 is probably too too small. Also, the labels Will not be intuitive to everyone, and should at least be explained in the legend

References are much overused. Once used, a reference doesnt have to keep being used unless it is introducing a new aspect (e.g. ref 31).

we noticed that for a group of proteins the first methionine from UniProtKB/SwissProt sequences differed from the first methionine indicated by our approach.”

It would be interesting to know which ones (supplementary, clearly). Also are SpliceProt sequences generally longer or shorter?

"canonical and non-canonical" 

What is the difference between the two here?

"OpenProt is currently the most well annotated databas"

Well, I imagine that it might be the largest, because it grabs everything and more. But I don’t think you could describe it as well annotated. Well annotated suggests that a lot of work has gone into matching features with proteins, rather than just button pushing.

"UniProtKB/SwissProt is the gold 127 standard database in shotgun proteomics analyses."

Is it? In that case there ought to be a reference. The users of Nextprot would certasinly dispute that claim (though it would be perfectly reasonable to point out that nextprot is basically a copy of UniProt). In fact, UniProt is well annotated. OpenProt might best be described as “extensive”.

The results of the comparison guided by Ensembl identifiers is a bit odd. SpliceProt vs. SwissProt ought to be 100% identical (one reason it isn’t is explained below), and SpliceProt vs OpenProt if guided by ensembl should not present so many differences, particularly if UniProt vs. OpenProt is so similar. This really needs further investigation.

The comparison of proteotypic peptides using SpliceProt / SpliceProt / OpenProt is not really clear. Nor is the purpose so obvious. I think I understand it more or less, but the authors are going to have to rewrite to (a) make more clear and (b) explain some of the frankly bizarre-looking results, eg, why are rat SpliceProt numbers so high and almost always the same? An introduction is important because it is explained better in the methods section afterwards.

The example of Glyctk in mouse is not a great example of AS because all four protein sequences are the same except that two of them are subsequences of the other two. The subsequences are an (unfortunate) side effect of using Ensembl. Align the four sequences to see that the peptides dont distinguish between the sequences.

When choosing a representative protein, it is a mistake to put TSL1 before APPRIS principal isoform. It isn’t so useful. The most functional interesting isoform in Ensembl will always be the MANE or APPRIS isoform see the papers by Pozo et al for detail. The only problem with this is that the new version of APPRIS may have been post-Ensembl100 and MANE Select transcripts certainly are.

Ensembl and UniProt work together and Ensembl and SwissProt sequences ought to be the same. However, Ensembl sequences are based on the reference genome, while UniProt sequences may be based on common variants to the reference genome. GENCODE/Ensembl is a conservative database and annotates sequences based on evidence. This makes it a good database for proteomics research. However, it has two disadvantages. The first is that it includes a lot of unfinished transcripts and therefore sequences (see Glyctk in mouse for example). The second is that it annotates readthrough transcripts as coding. This adds noise and reduces the number of proteotypic peptides because many peptides map to more than one gene. GENCODE/Ensembl is working to eradicate these problems, but it would be interesting to filter out the subsequences and readthroughs from SpliceProt in the meantime. Many readthrough transcripts are also tagged as NMD by GENCODE, but I don’t know whether the NMD prediction in SpliceProt would pick them up as such. Readthrough, and start and end not found tags are present in the GENCODE/Ensembl gtf.
